# Geographical Distribution, Host Range and Genetic Diversity of *Fusarium oxysporum* f. sp. *cubense* Causing *Fusarium* Wilt of Banana in India

**DOI:** 10.3390/jof10120887

**Published:** 2024-12-21

**Authors:** Raman Thangavelu, Hadimani Amaresh, Muthukathan Gopi, Murugan Loganathan, Boopathy Nithya, Perumal Ganga Devi, Chelliah Anuradha, Anbazhagan Thirugnanavel, Kalyansing Baburao Patil, Guy Blomme, Ramasamy Selvarajan

**Affiliations:** 1ICAR—National Research Center for Banana, Plant Pathology Division, Tiruchirappalli 620102, Tamil Nadu, India; amarhadimani.1773@gmail.com (H.A.); gopimusa@gmail.com (M.G.); logumuruga@gmail.com (M.L.); nithyanithyab14@gmail.com (B.N.); gnsganga@gmail.com (P.G.D.); anuradha.chelliah@gmail.com (C.A.); selvarajanr@gmail.com (R.S.); 2ICAR—Central Citrus Research Institute, Nagpur 440033, Maharashtra, India; lotus.thiru@gmail.com; 3Jain Irrigation System Ltd., Jalgaon 425002, Maharashtra, India; patil.kalyansing@jains.com; 4Bioversity International, c/o ILRI, Addis Ababa P.O. Box 5689, Ethiopia; g.blomme@cgiar.org

**Keywords:** *Fusarium* wilt, banana, geographical distribution, genetic diversity, host range, *Fusarium oxysporum* f. sp. *cubense*, SIX genes

## Abstract

*Fusarium* wilt of banana is a major production constraint in India, prompting banana growers to replace bananas with less remunerative crops. Effective disease management practices thus need to be developed and implemented to prevent further spread and damage caused by *Fusarium oxysporum* f. sp. *cubense* (*Foc*), the cause of *Fusarium* wilt. Currently, knowledge of disease incidence, affected varieties, and the geographical spread of *Foc* races in India are only scantily available. An extensive field survey was conducted in 53 districts of 16 major banana-growing states of and one union territory of India that covered both tropical and subtropical regions. Disease incidence ranged from 0 to 95% on farms, with Cavendish bananas (AAA) most affected. No *Fusarium* wilt symptoms due to *Foc* R1 were observed in Nendran (AAB) or Red Banana (AAA) in South India. During the survey, 293 *Foc* isolates were collected from Cavendish, Pisang Awak (ABB), Silk (AAB), Monthan (ABB), Neypoovan (AB), and Mysore (AAB) bananas. Isolate diversity was assessed through Vegetative Compatibility Group (VCG) analyses, sequencing of EF1α gene sequences, phylogenetic analyses, and characterisation by *SIX* gene composition. Thirteen VCGs were identified, of which VCGs 0124, 0125, 01220, and 01213/16 were dominant and infected Cavendish bananas. Phylogenetic analysis divided the Indian *Foc* isolates into race 1 (R1), subtropical race 4 (STR4), and tropical race 4 (TR4). *Secreted in Xylem* (*SIX)* gene analyses indicated that the effector genes *SIX*4 and *SIX*6 were present in the VCGs 0124, 0124/5, 0125, and 01220 of race 1, *SIX*7 was present only in *Foc* STR4, and *SIX*8 was found only in *Foc* R4 (TR4 and STR4) isolates. Insights into the geographical distribution of *Foc* races, and their interactions with banana varieties, can guide integrated disease management intervention strategies across India.

## 1. Introduction

Banana is a major fruit crop, with annual global production estimated at 135 million tonnes (MT). Of these, 24.3 MT are exported (https://www.nab.com.na/wp-content/uploads/2024/04/Market-Intelligence-Report-Bananas-revised-NAB-29032024.pdf, accessed on 4 November 2024). The global banana trade is worth USD 13.88 billion per year [1], making the commodity one of the most important in the world. Additionally, it is an important source of income, employment, and export revenue for developing countries in Latin America, Southeast Asia, and Africa [2]. In Asia, major banana-producing countries include India, China, and Indonesia (https://www.nab.com.na/wp-content/uploads/2024/04/Market-Intelligence-Report-Bananas-revised-NAB-29032024.pdf, accessed on 4 November 2024), where tropical climates favour their cultivation. Of the more than 1000 varieties of bananas known in the world, the Cavendish (AAA) variety contributes 47% to world banana production (https://www.fao.org/economic/est/est-commodities/oilcrops/bananas/bananafacts/en/, accessed on 4 November 2024). Cavendish banana is known to achieve high yields per hectare and, due to their short stems, are less prone to damage from environmental influences such as storms [3].

In India, banana ranks first of all fresh fruits produced, with a total production of 36.61 million tonnes cultivated on 0.996 million ha (https://www.indiastat.com/table/agriculture/selected-state-wise-area-production-productivity-b/1441986, accessed on 4 November 2024). Most bananas produced in the country are consumed locally, with only 0.36 million tonnes exported mainly to Iran, Iraq, UAE, Oman, Uzbekistan, Saudi Arabia, Nepal, Qatar, Kuwait, Bahrain, Afghanistan, and the Maldives ((https://pib.gov.in/PressReleaseIframePage.aspx?PRID=1976245, accessed on 4 November 2024). Bananas in India are grown in a range of climatic conditions, including the tropical southern states of Tamil Nadu, Kerala, Karnataka, Andhra Pradesh, Maharashtra, and Odisha, and the subtropical states in northern and north-eastern India (Madhya Pradesh, Uttar Pradesh, Bihar, West Bengal, Gujarat, Assam, Nagaland, Meghalaya, and Nagaland). The major banana-producing states include Andhra Pradesh (6.11 MT), Maharashtra (5.56 MT), Tamil Nadu (4.48 MT), Uttar Pradesh (3.39 MT), Gujarat (3.97 MT), Karnataka (2.55 MT), Madhya Pradesh (2.27 MT), Bihar (2.00 MT) and West Bengal (1.20 MT) (Source: Ministry of Agriculture & Farmers Welfare, Govt. of India, ON3447, accessed on 5 December 2024), while the key varieties grown are Gran Nain (AAA), Robusta (AAA), Dwarf Cavendish (AAA), Red Banana (AAA), NeyPoovan (AB), Nendran (Plantain-AAB), Poovan (Mysuru-AAB), Rasthali (Silk-AAB), Karpuravalli (Pisang Awak-ABB), and Monthan (Cooking banana-ABB). However, among these varieties, the Cavendish group of bananas is the major variety grown in the subtropical regions of India like Gujarat, Maharashtra, Madhya Pradesh, West Bengal, and Bihar states. In these locations, Cavendish is mainly grown as a monoculture crop.

Several biotic constraints affect banana production in India. These include black Sigatoka (or black leaf streak), caused by *Mycosphaerella fijiensis*, and *Fusarium* wilt, caused by *Fusarium oxysporum* f. sp. *cubense* (*Foc*) [4,5]. *Foc* is a soil-borne pathogen that produces resting spores, called chlamydospores, which help the pathogen survive for decades without its host [6]. Once *Foc* enters a banana field it cannot be eradicated, which hampers the future cultivation of susceptible banana varieties. The pathogen infects plants through the root system and colonises plant vascular tissues, thereby disrupting water and nutrient transportation in the xylem vessels of the rhizome and pseudostem [6]. This results in a reddish-brown discolouration of infected vascular tissue, which results in yellowing and browning of leaves, pseudostem splitting, and plant wilting [4,5]. Infected plants often die before flower emergence/shooting takes place. In heavily infested fields, symptoms are even produced on 1-month-old plants, forcing farmers to abandon their fields (personal observation).

Banana *Fusarium* wilt disease was first reported in Australia in 1874 and is now found in almost all the major banana-producing regions of the world [6]. *Foc*, the causal agent of *Fusarium* wilt, is divided into three races, based on the banana varieties it affects. These are *Foc* race 1 (R1), *Foc* race 2 (R2), and *Foc* race 4 (R4). *Foc* race 4 is further divided into subtropical race 4 (STR4) and tropical race 4 (TR4), depending on the climatic conditions where Cavendish bananas are affected [6]. The dessert banana variety “Gros Michel”, which once dominated world trade, was almost wiped out by *Foc* R1 in the 1900s, forcing trade to shift to the Cavendish subgroup, which was resistant to this race. However, in the 1990s, Cavendish bananas succumbed to *Foc* TR4 in Taiwan, Indonesia, Australia, and Malaysia [6]. *Foc* TR4 causes wilt in Cavendish bananas under both tropical and sub-tropical conditions and affects more varieties than any other *Foc* race. In India, Cavendish bananas were also reported to be affected by *Foc* R1, both in tropical and subtropical regions [7,8]. This rarity was first reported in the Theni district of Tamil Nadu in 2010, but it subsequently spread to Gujarat and is now found in most banana-growing regions of India [9].

*Foc*TR4 was first isolated from Robusta (AAA) and Grand Nain (AAA) bananas in the Katihar district of Bihar State in India in 2015 [10]. The pathogen has since spread to neighbouring states like Uttar Pradesh, West Bengal, Gujarat, Maharashtra, and Madhya Pradesh. Heavy yield losses caused by *Foc* TR4 in Cavendish-producing regions forced growers to plant less profitable crops like maize and paddy in infested fields, which has negatively affected employment opportunities for banana workers (unpublished data). As both *Foc* R1 and TR4 might be spreading rapidly in northern India, Cavendish production on more than 4 million ha, worth INR 30,000 to 40,000 crores (USD 48 million), is at risk.

There is an urgent need to develop and implement management practices to mitigate *Fusarium* wilt and prevent the further spread of *Foc* in India. Information on the geographical occurrence and incidence of *Fusarium* wilt, and the distribution of *Foc* VCGs/races, is crucial for effective decision-making and timely deployment of available management practices. Thus, the present study conducted field surveys in major banana-growing regions of India to gain insight into the extent of the spread of the disease and to know the distribution of different *Foc* races and VCGs of the pathogen, all aiming to prevent the pathogen’s spread and effective management of this deadly disease.

## 2. Materials and Methods

### 2.1. Survey on Fusarium Wilt Disease

A roving survey was conducted on 286 farms in 53 districts and 16 banana-growing states and one union territory of India from 2019 to 2023 to assess the incidence of banana *Fusarium* wilt and to collect *Foc* samples for the isolation and characterisation of the pathogen (Table 1). Banana varieties grown in both tropical and subtropical regions were covered in the survey. In each farm surveyed, a minimum of 500 to 1000 plants were examined for disease symptoms, and at least two to three samples were collected per farm for the isolation of the *Foc* pathogen. Farmers were also interviewed to gather information about the farm: cultivar(s) grown, source of planting material (tissue cultured/suckers), current management practices (chemical/biological/cultural practices like disinfestation of tools and implements, removal and destruction of infected plants), type of irrigation used (drip/flood), quarantine measures, and crop rotation practices. The goal of the questionnaire was to determine modes of disease spread and management practices being applied by farmers, which could aid in designing effective future management strategies for *Fusarium* wilt in India.

### 2.2. Sample Collection and Pathogen Isolation

During the field survey, plants showing symptoms of *Fusarium* wilt, such as leaf yellowing and pseudostem splitting, were cut down and examined for vascular discolouration in the rhizome and pseudostem. Pieces of discoloured vascular tissue from both rhizome and pseudostem were collected and dried on sterile filter paper. The filter papers were changed daily until these were brought to the laboratory at ICAR-National Research Centre for Banana (NRCB), Tiruchirapalli, for the isolation of the pathogen.

At the laboratory, 5–8 mm long dried diseased tissue sections were excised from the discoloured vascular strands and surface sterilised, and four dried pieces of each sample were plated onto plates of quarter-strength potato-dextrose agar (PDA). After 3 days, isolates of *F. oxysporum* were sub-cultured onto PDA plates and incubated at 25 °C. Single spore (monoconidial) cultures were generated after 7 to 10 days by using a dilution plating technique [11]. The cultures were then stored on PDA and dried filter papers at 4 °C for short-term storage, and in 15% glycerol at −80 °C for long-term storage.

### 2.3. Morphological Identification

The *Fusarium* cultures isolated from diseased banana tissues were grown on PDA and Carnation Leaf Agar plates at 25 °C for 10 days. Isolates were identified to species level based on their colony colour and the morphological characteristics [12] of macro and micro conidia and chlamydospores.

### 2.4. Pathogenicity Testing

The pathogenicity of each *F. oxysporum* isolate was tested by inoculating 3-month-old tissue-cultured banana plants (Grand Nain, NeyPoovan, Poovan, Monthan, Rasthali and Karpuravalli) obtained from Jain Irrigation Pty. Ltd. (Udumalpet, Coimbatore, India) with the isolated fungi. The individual isolates were first multiplied in a sand–maize (19:1) medium (briefly, 190 g of river sand and 10 g of maize flour were taken in a plastic container, and 20 mL of water was added and mixed well; the mixture was taken in an autoclavable bag and sterilised by autoclaving at 15 lb for 20 min, 2 times at a one–day interval). The sand–maize meal medium was inoculated with a piece of actively growing fungal culture and incubated at 30 °C for 15 days, and inoculations were performed on 3-month-old potted TC plants by applying 30 g of inoculums to each plant at the sub-surface soil [13]. Five plants were inoculated with each isolate and the pots were arranged in a completely randomised design. Five plants grown in a non-inoculated sand:maize meal mixture served as control. The plants were then maintained in a greenhouse at 25 °C with a 12 h photoperiod. The inoculated plants were inspected for external and internal *Fusarium* wilt symptoms 3 months after inoculation, and rhizome discolouration was rated on a 0–5 scale using the method of Zuo et al. [14]. The fungus was re-isolated from the pseudostem of diseased plants and identified by PCR using race-specific *Foc* primers [15].

### 2.5. Vegetative Compatibility Group Analysis

Chlorate-resistant, nitrate-non utilising (*nit*) mutants were developed for all the *Foc* isolates collected in India, and phenotyped, as described by Correll et al. [16]. The unknown *nit*-mutants were then paired with Nit-M tester strains obtained from the Department of Agriculture and Fisheries, Queensland, Australia, using the method described by Thangavelu et al. [17]. Briefly, a mycelial disc (2 mm in diameter) of a known Nit-M tester was placed in the centre of a Petri dish containing minimal medium, and small *nit*-1 mutant disks of an unknown VCG were placed 10–15 mm away. The plates were then incubated at room temperature (24 ± 2 °C) and examined regularly for the formation of wild-type growth (heterokaryons) at the line of contact between the two colonies. Isolates that formed heterokaryons were assigned to the same VCG.

### 2.6. Cross Infection Studies

To determine whether *Foc* R1 isolates, particularly those belonging to VCGs 0124, 0125, 0124/5, and 01220, isolated in this study from cultivars other than Cavendish could also infect Cavendish bananas, a cross-infection study was conducted under glasshouse conditions using Grand Nain. For this, 3-month-old disease-free tissue-cultured banana plants of the cv. Grand Nain were inoculated with *Foc* R1 isolates as described before, with 10 replications per isolate. After 3 months, the plants were rated for internal rhizome symptoms on a scale of 0 to 5 as described earlier [14].

### 2.7. Molecular Identification of Foc Isolates

#### 2.7.1. Extraction of Fungal Genomic DNA

Five-day-old *Foc* cultures were inoculated into potato-dextrose broth (PDB) amended with streptomycin sulphate (0.1 g L^−^^1^). The cultures were subsequently incubated at 25 °C with a 12 h alternating light and dark cycle for 5 days. After incubation, the mycelial mat from the broth was removed for DNA extraction [18] using the method of Raeder and Broda [19] with some modifications. Briefly, the powdered mycelium was suspended in 800 µL of extraction buffer (200 mM Tris-HCL pH 8.5, 25 mM NaCl, 25 mM EDTA, 0.5% SDS) and incubated for 2 h at 37 °C. After incubation, the tubes were centrifuged at 10,000 rpm for 10 min. The supernatant alone was transferred to a new tube and an equal volume of phenol–chloroform–isoamyl alcohol (25:24:1) was added. The tubes were then again centrifuged at 10,000 rpm for 10 min. The aqueous phase was transferred to a new tube and an equal amount of phenol–chloroform–isoamyl alcohol (25:24:1) was added. The tubes were again centrifuged at 10,000 rpm for 10 min. DNA was precipitated with isopropanol and 3 M sodium acetate by incubating the tubes at −20 °C overnight. The DNA pellet was rinsed with 70% ethanol and re-suspended thoroughly in TE buffer (10 mM Tris-HCL, 1 mM EDTA, pH8.0). The purification was performed by incubating the isolated DNA with 5 µL RNAse A at 37 °C for 1 h to remove RNA. The extraction was re-suspended in a TE buffer and stored at −20 °C until further use. The quality and concentration of DNA were determined using a DU640 spectrophotometer (Lambda 25, PerkinElmer, and Norwalk, CT, USA) and visualised on 0.8% agarose gel. The concentration of DNA was adjusted to 25 ng µL^−1^.

#### 2.7.2. Identification of Foc Races by PCR

DNA amplification was performed in a Master cycler nexus gradient PCR machine (Eppendorf India PVT. LTD., Chennai, India) using *Foc* R1-, TR4-, and STR4-specific markers (Table 2) developed and described by Thangavelu et al. [15]. Briefly, 20 µL of the PCR reaction mixture containing forward and reverse primers (0.5 µM µL^−1^ each), EmeraldAmp^®^ PCR master mix (9.0 µL), ultra-pure water (8 µL), and 50 ng µL^−1^ DNA was used for PCR amplification. PCR was performed using the following conditions in a thermocycler (Takara, Shiga, Japan): denaturation at 95 °C for 5 min followed by 30 cycles of amplification at 95 °C for 30 s, annealing at 65 °C for 40 s and 72 °C for 45 s, and final elongation at 72 °C for 5 min; in the case of *Foc* TR4-, R4-, and STR4-specific amplification, annealing was carried out at 66 °C for 40 s. Electrophoresis was undertaken using 2% agarose gel at 100 V for 30 min and visualised on the gel documentation system under ultraviolet light (302 nm).

#### 2.7.3. Sequencing of TEF-1α Gene and Phylogenetic Analyses

The translation elongation factor-1α (TEF-1α) gene of 46 representative *Foc* isolates belonging to major VCGs that have been infecting most of the banana varieties, including Cavendish cultivars, and frequently encountered in most of the banana-growing regions in India, such as VCGs 0124, 0124/5, 0125, and 01220 of *Foc* R1, VCG 0120 of *Foc* STR4, and VCG 01213/16 of *Foc* TR4, were amplified with primers EF-1 and EF-2 as follows. For PCR, a reaction volume of 40 μL was prepared consisting of 1 unit of Taq polymerase (BIOTAQ, UK), 1× PCR buffer, 3.5 mM MgCl2, 200 μM of each dNTP, bovine serum albumin (BSA), 0.2 μM of each primer, and genomic DNA (50 ng), and the PCR program as set at 95 °C for 2 min followed by 35 cycles of 95 °C for 30 s; 50 °C for 30 s; and 72 °C for 1 min, and an additional extension time for 10 min at 72 °C. The PCR products were separated by electrophoresis in a 1% agarose gel. The PCR products were then purified using the DNA Gel Extraction kit TSP601, and sequenced from both ends using an ABI3730 XL DNA analyser (Genurem Biotech Ltd., Bengaluru, India). The TEF-1α sequences of a *Foc*TR4 isolate from Malaysia and two isolates of *Fusarium equiseti* (as an outgroup) were retrieved from the NCBI website (https://www.ncbi.nlm.nih.gov/, accessed on 2 November 2024). The TEF-1α gene sequences of all *Fusarium* isolates were aligned with clustalW, and the phylogenetic analyses performed using the maximum likelihood method. The phylogenetic tree was constructed using MEGA 11 software with the Kimura 2-parameter model. Branches were tested for the inferred tree by bootstrap analysis on 1000 random trees.

#### 2.7.4. Characterisation of Foc Isolates by SIX Genes Amplification

The primers of *SIX* genes specifically designed for Indian *Foc* isolates were used in this study [20]. Also, in earlier studies, it was found that out of 14 *SIX* effector gene primers (*SIX1***–***SIX14*), all the *SIX* primers except *SIX3*, *SIX5*, *SIX10*-*SIX12,* and *SIX14* generated the expected amplicon from the Indian *Foc* isolates [20]. Hence, in this study, only eight different *SIX* primers (*SIX1*, *SIX2*, *SIX4*, *SIX6-SIX9*, and *SIX13*) were used for characterisation of *Foc* isolates (). The presence/absence of *SIX* effector genes in Indian *Foc* was determined using isolates representative of VCGs in *Foc* R1, TR4, and STR4 collected in different banana-growing regions of India. The *SIX* gene-PCR reaction mixture was set up in a total volume of 15 µL containing forward and reverse primers (1 µM µL^−1^ each), Emerald Amp^®^ PCR master mix (7.5 µL), ultra-pure water (3.5 µL), and 2 µL of 50 ng µL^−1^ DNA. The PCR for the *SIX* genes was set up with standard thermocycling conditions: initial denaturation at 95 °C for 5 min, 30 cycles of denaturation at 95 °C for 30 s, annealing at 59 °C/60 °C for 45 s, extension at 72 °C for 45 s, followed by one cycle of final extension at 72 °C for 10 min (Table 2). Electrophoresis was undertaken using 2% agarose gel at 100 V for 30 min, and visualised on the gel documentation system under ultraviolet light (302 nm).

## 3. Results

### 3.1. Distribution and Incidence of Fusarium Wilt

The *Fusarium* wilt symptoms were observed in Cavendish (Grand Nain, Singapuri and Robusta), Rasthali-Silk (Malbhog, Mortaman, Amritapani, Nanjangod Rasabale), Poovan-Mysuru (Cheni champa, Alpon), Karpuravalli (Chinia, Manohar, Kanthali), NeyPoovan, and Athiakol, but not in Nendran or Red Banana, which are grown only in Tamil Nadu and Kerala where *Foc* R1 is omnipresent and TR4 is currently absent. In general, the disease incidence was ranged from 0 to 95%, with maximum incidences of 90–95% observed in Grand Nain grown in the Theni district of Tamil Nadu (tropical) and the Lakhimpur-Keri district of Uttar Pradesh (sub-tropical). This was followed by the varieties Poovan (West Bengal) and Rasthali (Tamil Nadu), which recorded a 45–50% incidence in both tropical and subtropical regions of India (Table 1).

A total of 732 *Fusarium* wilt-affected samples were collected, from which 293 representative *Fusarium* isolates were obtained for further characterisation. Among these, 51.2% were isolated from Cavendish cultivars, 18.08% from Rasthali, 10.88% from Karpuravalli, 9.52% from Monthan, 5.78% from NeyPoovan, 3.4% from Poovan, and 1.02% from other varieties (Table 3). The interview with farmers revealed that more than 60% of the farmers in India use suckers as planting material and follow flood irrigation. Unfortunately, in most of the cases, no management practices were followed including quarantine and crop rotation.

### 3.2. Pathogenicity and Cross Infection Studies

Banana plants of respective varieties inoculated with *Fusarium* isolates (293 isolates) revealed typical symptoms of *Fusarium* wilt such as leaf yellowing and chlorosis in 30–40 days, and whole plant wilting occurred mostly in 80 to 90 days. Internal rhizome discolouration had scores ranging from 4 to 5 on a 0–5 scale.

Cross-infection studies also indicated that VCGs 0124, 0125, 0124/5, and 01220 caused internal disease scores of 3 to 5 on a 0–5 disease scale in Cavendish cv Grand Nain plantlets (Figure 1). The pathogen was re-isolated from infected portions of the rhizome and was reconfirmed by using molecular markers specific to *Foc* R1 and *Foc* TR4.

### 3.3. Morphological Identification

The mycelia of all *Fusarium* cultures were cottony, white to pale violet, and produced dark red or pale violet colour pigments in PDA. They produced numerous one- or two-celled, oval micro conidia in false heads. The four- to eight-celled macro conidia were sickle-shaped with foot-shaped basal cells. Chlamydospores were globose and formed singly or in pairs. Based on these morphological characters, the *Fusarium* isolates were identified as *F. oxysporum*.

### 3.4. VCG Analyses

Thirteen *Foc* VCGs and VCG complexes were identified from isolates collected in India. These were VCGs 0124, 0124/5, 0125, 0128, 01220, 01212, 01214, 01217, 01218 in *Foc* R1, VCGs 0120, 0129, 01211 of *Foc*STR4, and VCG 01213/16 of *Foc* TR4 (Table 4). Among the cultivars grown, Rasthali was associated with the highest number of VCGs (10), followed by Karpuravalli (7), NeyPoovan (6), and Grand Nain (5). Poovan and Athiakol were associated with two VCGs each (Figure 2). *Foc* TR4 VCG 01213/16 was isolated from varieties such as Cavendish (Grand Nain, Robusta, and Singapuri), Karpuravalli, and Poovan in subtropical regions of India. *Foc* R1 VCGs 0125, VCG 0124, and 01220 affected most banana varieties in India, including Cavendish bananas, both in tropical and subtropical regions of India. *Foc* STR4 VCG 0120 affected Cavendish bananas in the Gujarat and Madhya Pradesh states in subtropical regions of India, whereas VCG 0129 was associated with Rasthali bananas in the subtropical state of Assam and the tropical state of Andhra Pradesh. VCG 01211 affected NeyPoovan in Karnataka state (tropics) and Karpuravalli in Assam state (subtropics) (Table 4 and Figure 3).

### 3.5. Molecular Identification

#### 3.5.1. Identification of Foc Races by PCR

The primer sets *Foc* R1F and *Foc* R1R, which are specific to *Foc* R1, *Foc* STR4F and *Foc*STR4R, which are specific to STR4, and *Foc*R4F and *Foc*R4R, specific to *Foc*R4, and the primer sets *Foc*TR4F and *Foc*TR4R, which are specific to TR4, yielded the expected amplificons with the size of 320 bp, 250 bp, 400 bp, and 250 bp, respectively, for *Foc* isolates belonging to VCGs in *Foc* R1, STR4, and TR4 (Table 5).

#### 3.5.2. TEF-1α Gene Sequencing and Phylogenetic Analyses

PCR amplification of the TEF1-α gene produced ~690 bp fragments for sequencing. The TEF1-α gene sequences of 46 *Foc* isolates were deposited into the GenBank repository (accession number MN867468; MW286790–91, 93, 96; MW286802–04, 06, 10; MW339762, 63, 66, 67, 70–73, 75, 77, 79–81; OR468255–57, 61–63, 67–79, 82–85). A phylogenetic tree generated from the *Foc* sequences indicated two major clades, A and B (Figure 4). Clade A contains all the *Foc* R1 isolates, and clade B contains all the *Foc* R4 isolates, which include TR4 and STR4 isolates. The TR4 and STR4 isolates in clade B were also clearly separated. The outgroup members of *F. equiseti* (MZ669768 and KX463032) were also separated from all the *Foc* isolates. The genetic similarity of Indian *Foc* isolates ranged from 97.4 to 100%, while the genetic dissimilarity between the outgroups *F. equiseti* and *Foc* isolates ranged from 41.4 to 49.2%. The genetic similarity between the *Foc* TR4 of China and Malaysia isolates, and the Indian *Foc* TR4 isolates ranged from 96.8 to 99.8%, and between the *Foc* R1 isolates of China and Central Africa, and the Indian *Foc* R1 isolates ranged from 97.6 to 99.8%. In this study, interestingly there was a clear separation of all the Indian *Foc* isolates into three major groups based on the races *Foc* R1, *Foc* STR4, and *Foc* TR4, irrespective of the geographical origin. However, the genetic similarity among the Indian *Foc* R1, STR4, and TR4 isolates ranged from 99.2 to 100%, 99.8 to 100%, and 99.5 to 100%, respectively. Genetic dissimilarity between *Foc* R1 and *Foc* TR4, which both infected most of the commercial cultivars, including Cavendish, ranged from 0 to 4.4% (Figure 4 and Appendix A).

#### 3.5.3. SIX Gene Analyses

The PCR amplification of *SIX* genes of Indian representative *Foc* isolates indicated that the expected amplicon sizes of effector genes *SIX4* (717 bp) and *SIX6* (412 bp) were generated for VCGs 0124, 0124/5, 0125, and 01220 of *Foc* R1, with *SIX7* (365 bp) being amplified only in *Foc* STR4, and SIX 8 (450 bp) amplified only in *Foc* TR4 isolates. The effector genes *SIX1*, *SIX9*, and *SIX13* were amplified in all the VCGs, whereas *SIX2* (214 bp) was amplified in both *Foc* STR4 (VCG0120) and *Foc* TR4 (VCG01213/16) (Table 6 and Figure 5).

## 4. Discussion

*Fusarium* wilt is a major threat to banana cultivation in India. A *Fusarium* field survey conducted in 2019 showed lower infection rates: 6–65% in Bihar, 30–45% in Uttar Pradesh, and 5–15% in Gujarat in the subtropical regions, and 15–21% in Tamil Nadu in the tropical region [8]. Similarly, in the Jalgaon district of Maharashtra, which is considered the “Banana city of India” as it produces 0.43 million tonnes of banana from 0.062 million hectares, the incidence in 2019 was only 2% [21]. However, during the current 2024 survey, a 30% field level incidence was observed. These observations point to a fast pathogen/disease spread over the past 5 years. The major banana varieties affected by *Fusarium* wilt disease in the latest survey were Cavendish, Silk, Cooking type (ABB), Pisang Awak, Poovan, and NeyPoovan. However, among different varieties, the Cavendish bananas were heavily affected by *Fusarium* wilt, with infection rates ranging from 2 to 95% in tropical regions (maximum of 90% in Tamil Nadu), and from 0.5 to 90% in subtropical regions. Various reasons for the observed sharp increase in disease incidence and geographical spread emerged during the banana farmer interviews: (i) continuous cultivation of banana without any crop rotation as the farmers receive high profits from banana (USD 2500 to 3500 from banana as against USD 300 to 600 from maize/paddy rice/ sugarcane); (ii) lack of awareness of the modes of spread of the pathogen and eventual impact; (iii) no application of measures to prevent pathogen spread or to mitigate disease impact; (iv) use of suckers as planting material sourced from diseased mats or fields; and (iv) application of flood irrigation instead of drip irrigation. Farmers from the Baruch district of Gujarat expressed that severe flooding, caused by the overflow of water from the Narmada River, led to an extensive spread and high incidence of *Foc* TR4. The same means of spread was reported in other studies [22,23,24,25,26]. The high disease incidence in India resulted in most farmers planting resistant banana varieties like Nendran (AAB) or Red Banana (AAA), particularly in Theni district of Tamil Nadu, where *Foc* R1 is the major problem in Cavendish, or other crops like potato, maize, or sugarcane in Bihar, Uttar Pradesh, and West Bengal, where TR4 is the major problem and affects all cultivars.

Most of the *Foc* isolates collected in India in this study were from Cavendish bananas (51.2%). This may be because approximately 57% of the banana-growing areas in India are occupied by the Cavendish group [10]. The isolates were not only members of *Foc* TR4 (VCG 01213/16), as would be expected, but also *Foc* R1 VCGs, including VCG 0124, 0125, 0124/5, and 01220. The *Foc* R1 strains also infected Cavendish *cv.* Grand Nain under glasshouse conditions, as was previously reported by Thangavelu et al. [8,17,27]. Interestingly, among the VCGs, VCGs 0124, 0125, and 01220 of *Foc* R1, which were initially recorded in Cavendish bananas grown in the Theni district of Tamil Nadu [8,9], later spread to Uttar Pradesh and Gujarat. These VCGs are now found to affect all commercial cultivars grown in other major banana-producing states of India, including Cavendish (but not Nendran and Red Banana, which are grown only in southern India), and are encountered more frequently as well [9]. Regarding VCG 01213/16 (TR4), which was initially recorded in Katihar and Purnia districts of Bihar [10], the present study has revealed its further spread to other major banana-growing neighbouring states like Maharashtra (Jalgaon district), Gujarat (Surat and Bhruch districts), Uttar Pradesh (Maharajkanj,, Kushi Nagar, Shravasti, Ayodhya, Lakhimpur-Kheri, and Barabanki), Madhya Pradesh (Burhanpur), Bihar (Bhagalpur), and West Bengal (Nadia and Mushidabad). It is obvious from the present study that in addition to VCG 01213/16, the fast spread of *Foc* R1 VCGs (VCG 0124, 0125, and 1220) increasingly threatens Cavendish production landscapes. Therefore, understanding the banana–*Foc* race interaction is essential for the development and deployment of resistant bananas, and for other disease management strategies such as the development of effective biocontrol strains.

VCG analysis is a key method used for diversity analysis of *Foc*, which is crucial for understanding the diversity both between races and within races of this pathogen [16,28]. In the present study, a total of 13 different VCGs were observed and among these, 8 VCGs, viz., 124, 0125, 0128, 0129, 01211, 01212, 01220, and 0124/5, were identified in both tropical and subtropical regions. In contrast, VCGs 01214 and 01217 were identified exclusively in tropical regions, while VCGs 0120, 01216, 01218, and 01213/16 were found only in subtropical regions. Around 20 different varieties of bananas are grown commercially across 0.96 million hectares in various climatic regions, both tropical and subtropical, in India. This diversity in banana cultivation, across a wide geographical region, may be one of the reasons for the great number of VCGs found in the country. Li et al. [29] conducted similar studies on the diversity and distribution of *Foc* in China and found that *Foc* was highly diverse, with a total of 11 VCGs identified. Of these, six VCGs (0120/15, 0123, 012313/16, and 01221) were found in Cavendish bananas. They also observed that VCG 01213/16 (TR4) was the major VCG affecting Cavendish bananas in Guangdong province, located in the southern subtropics. The identification of these VCGs and their locations is valuable for implementing quarantine measures and integrated disease management (IDM) practices, including the timely deployment of resistant varieties.

The phylogenetic tree grouped *Foc* isolates in India into three major groups as per the races *Foc* R1, *Foc* STR4, and *Foc* TR4, irrespective of their geographical origin and variety from which they were isolated. It also indicated the polyphyletic nature of Indian *Foc* isolates, as there was a wide genetic diversity among the *Foc* isolates of each race of India, which were all found to infect the Cavendish group of cultivars. Similar studies conducted previously by many researchers indicated that the *Foc* isolates have a wide genetic diversity and are polyphyletic in nature [17,30,31,32,33]. The phylogenetic tree results were supported by results obtained with a set of *SIX* genes, which verifies host specificity in the banana *Fusarium* wilt pathogen [34]. Czislowski et al. [35] demonstrated that VCGs of *Foc* races 1 and 2 were distinguishable from the VCGs of race 4 by both the presence and absence of several *SIX* genes. They indicated that *SIX2* (all R4 VCGs except 0122 and 01212) and *SIX8* (*SIX8a* in R4 and *SIX8b* in STR4) were present only in *Foc* R4, and SIX7 was present only in STR4, while SIX 1, 4, 6, 9, and 13 were present in all *Foc* races. It was also observed in the present study that the SIX genes such as *SIX1*, *SIX9,* and *SIX13* were present in all the VCGs of *Foc* R1, STR4, and TR4. In contrast, Czislowski et al. [35] stated that, of the seven *SIX* genes detected in the VCGs of *Foc*, only *SIX1* and *SIX9* were identified in all VCGs. However, the SIX 13 gene was not identified in all VCGs. In accordance with our findings, An et al. [36] observed that *SIX1*, *SIX8,* and *SIX9*, which are present in *Foc* TR4, are involved in the infection and colonisation of banana plants. In addition, in the present study it was found that the *SIX2* and *SIX8* were present only in *Foc* R4. In line with these findings, An et al. [36] showed that *SIX2* and *SIX8* effector genes were only present in *Foc* R4 and were not detected in *Foc* R1. Another study [20] carried out on multiple sequence alignment of extracted *SIX1* gene segments revealed significant variations between the VCGs of Indian isolates and those isolated by Czislowski et al. [35] and Guo et al. [37], specifically in the aligned positions of 207–230 and 535–552. This variation observed in the *SIX* gene sequences might be the reason why Indian *Foc* isolates, particularly the VCGs 0124, 1025, 0124/5, and 01220 of *Foc* R1, are infecting Cavendish bananas in India.

## 5. Conclusions

The present study highlights that *Fusarium* wilt disease caused by *Foc* R1 and TR4 is severely affecting bananas, especially the Cavendish variety, in India. *Foc*TR4, initially recorded in Bihar and Uttar Pradesh, has now spread to other major banana-growing states such as West Bengal, Gujarat, Maharashtra, and Madhya Pradesh. A total of 13 different vegetative compatibility groups (VCGs) were recorded, with VCGs 0125, 0124, 01220, and 01213/16 being dominant and infecting all banana varieties, including Cavendish. Interestingly, the Nendran and Red Banana varieties were not infected by any *Foc* race 1 strains, suggesting that these two varieties could be promoted in *Foc* race 1-infested areas. Phylogenetic diversity analysis using TEF1-α gene sequences and SIX gene analyses on representative *Foc* isolates clearly distinguished the races as *Foc R*1, STR4, and TR4. The findings from this study will be instrumental in the design, development, and timely deployment of integrated disease management (IDM) practices, including the use of resistant varieties.

## Figures and Tables

**Figure 1 jof-10-00887-f001:**
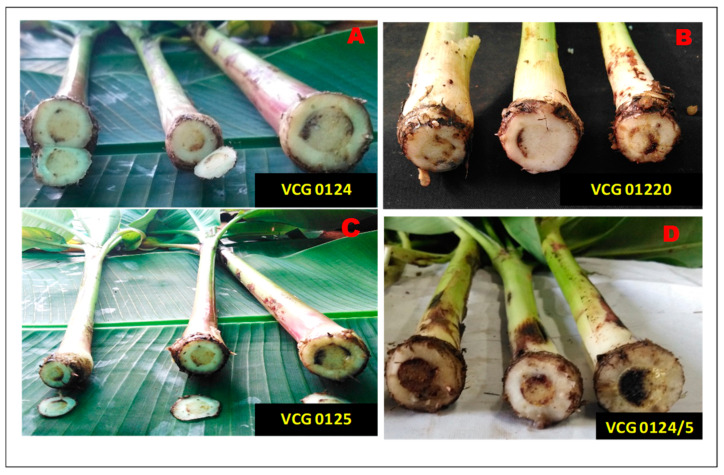
VCG’s of *Fusarium oxysporum* f. sp. *cubense* race 1 causing infection of Cavendish cv. Grand Nain plantlets in a glass house. (**A**) Cross-section of rhizome infected by VCG0124; (**B**) VCG 01220; (**C**) VCG0125 and (**D**) VCG0124/5.

**Figure 2 jof-10-00887-f002:**
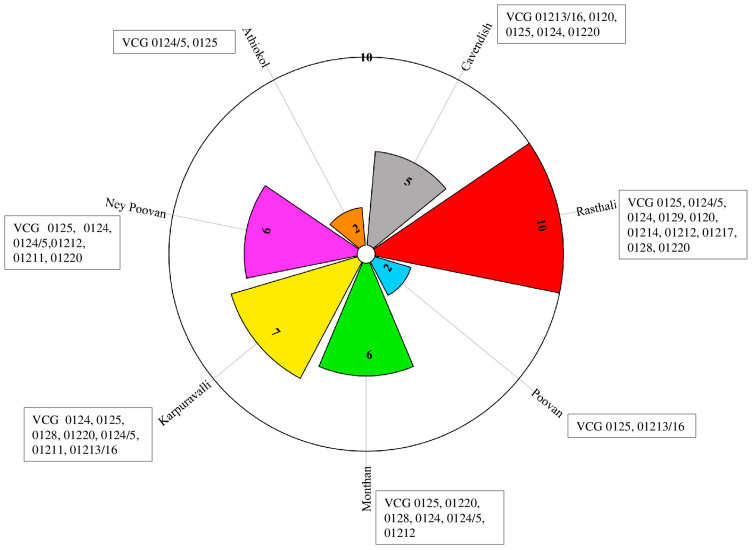
Number of *Fusarium oxysporum* f. sp. *cubense* VCGs associated with major commercial bananas grown in India.

**Figure 3 jof-10-00887-f003:**
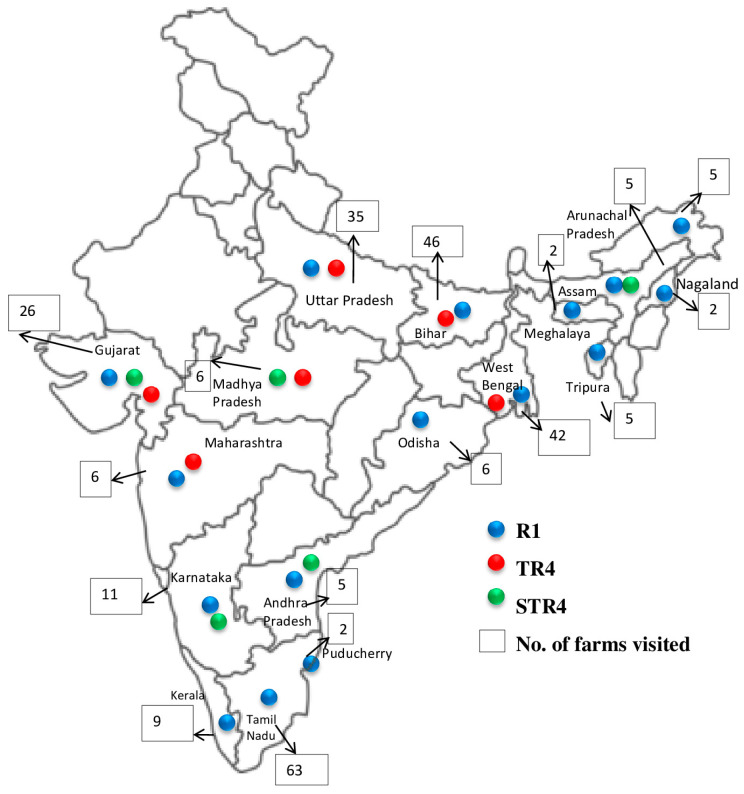
Geographical distribution of *Fusarium oxysporum* f. sp. *cubense* races in different banana-growing states of India.

**Figure 4 jof-10-00887-f004:**
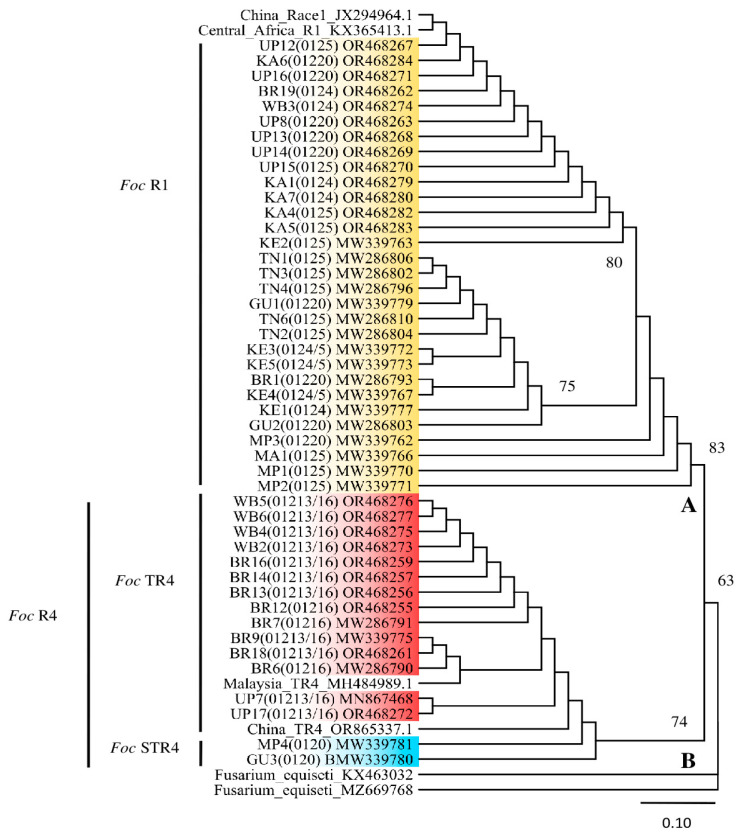
Phylogenetic tree constructed using the maximum likelihood method based on the TEF-1α gene sequences data of 46 representative *Fusarium oxysporum* f. sp. *cubense* (*Foc*) isolates associated with banana grown in India. The analysis was carried out with the TEF-1α sequences of Indian *Foc* isolates combined with sequences from *F. equiseti* (MZ669768.1 and KX463032), *Foc* TR4 of China (OR865337) and Malaysia (MH 484989), and *Foc* R1 of Central Africa (KX365413) and China (JX294964). Bootstrap values greater than 60% are indicated for maximum likelihood internodes where relevant. The scale bar corresponds to 0.10 nucleotide substitutions per site. The tree is rooted with two isolates of *Fusarium equiseti*.

**Figure 5 jof-10-00887-f005:**
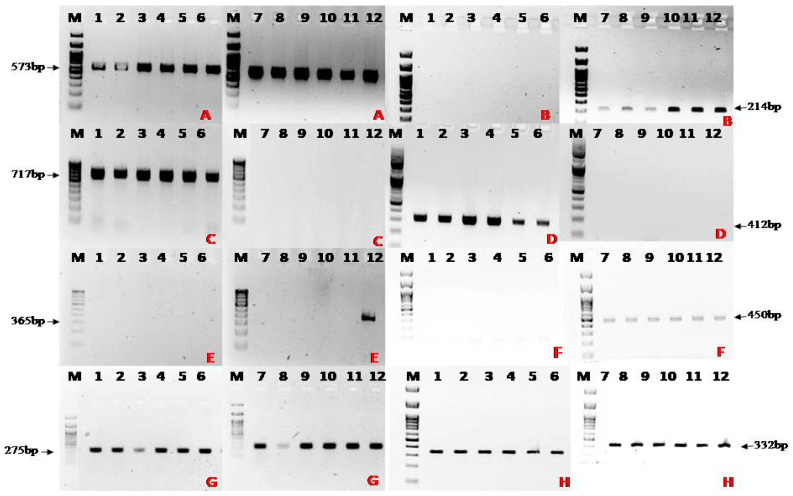
Distribution of effector-based *SIX* genes in different strains of *Fusarium oxysporum* f. sp. *cubense* (*Foc*) in India: (**A**) amplicons of SIX1F/SIX1R primer set for *Foc* R1, *Foc* TR4, and *Foc* STR4; (**B**) amplicons of SIX2F/SIX2R primer set for *Foc* R1, *Foc* TR4, and *Foc* STR4; (**C**) amplicons of SIX4F/SIX4R primer set for *Foc* R1, *Foc* TR4, and *Foc* STR4; (**D**) amplicons of SIX6F/SIX6R primer set for *Foc* R1, *Foc* TR4, and *Foc* STR4; (**E**) amplicons of SIX7F/SIX7R primer set for *Foc* R1, *Foc* TR4, and *Foc* STR4; (**F**) amplicons of SIX8F/SIX8R primer set for *Foc* R1, *Foc* TR4, and *Foc* STR4; (**G**) amplicons of SIX9F/SIX9R primer set for *Foc* R1, *Foc* TR4, and *Foc* STR4; (**H**) amplicons of SIX13F/SIX13R primer set for *Foc* R1, *Foc* TR4, and *Foc* STR4. Lane M: Ladder; Lane 1–6: *Foc* R1(VCG 01220, 0124/5, 0124, 0125); Lane 7–11: *Foc* TR4(VCG 01213/16); Lane 12: *Foc* STR4(VCG 0120).

**Table 1 jof-10-00887-t001:** Geographical distribution of different VCGS and races of the *Fusarium* wilt pathogen, and disease incidence according to banana variety.

State	District	Number of Plantations Visited	Varieties (Genomic Group)	Disease Incidence Range (%)	VCGs Identified	Races Identified
Tropical regions						
Tamil Nadu	Tuticorin	10	Monthan (ABB)	5–23	0125, 01220, 0128, 0124, 0124/5, 01220/01212	R1, R2
	Tirunelveli	4	Monthan (ABB)	7–25	01220	R1
		3	Rasthali (AAB)	3–40	0125	R1
		3	NeyPoovan (AB)	1–15	0125, 0124/5, 0124, 01220	R1
	Madurai	5	Monthan (ABB)	3 to 20	01220, 0125	R1
		2	Rasthali (AAB)		0125, 0124/5	R1
	Kanyakumari	8	NeyPoovan (AB)	8–13	0125, 0124	R1
	Namakkal	11	Rasthali (AAB)	23	0124, 0125, 01214	R1
	Theni	10	Grand Nain (AAA)	8 to 95	0125, 0124, 01220	R1
	Coimbatore	3	Karpuravalli (ABB)	0.5 to 13	0124, 0125, 0128	R1
	Karur	1	Karpuravalli (ABB)	5	0125, 01220	R1
		2	Rasthali (AAB)	5–15	0125	R1
	Thanjavur	1	Karpuravalli (ABB)	6	0124/5	R1
Pondicherry (UT)	Pondicherry	2	Karpuravalli (ABB)	5–9	0124	R1
Kerala	Thrissur	6	Rasthali (AAB)	11–25	0124/5, 01212	R1
	Palakad	1	NeyPoovan	24	0125	R1
	Trivandrun	2	Karpuravalli (ABB)	2–9	0128, 0124	R1
Karnataka	Mandya	2	Grand Nain (AAA)	2	0124	R1
		1	Rasthali (AAB)	25	0125	R1
	Chikkeballapura	3	NeyPoovan (AB)	30	0125, 0124/5, 01212	R1
	Mysuru	3	Nanjangud Rasabale (AAB)	0.4 to 30	0125	R1
		2	NeyPoovan (AB)	20–30	01220, 0125, 01211	R1, STR4
Andhra Pradesh	West Godavari	4	Mortaman/ Amritapani (AAB)	5–45	0124, 0125, 0129 01217	R1, STR4
	Guntur	1	Karpuravalli (ABB)	8	0124	R1
Odisha	Cuttak	1	Rasthali (AAB)	20	0125	R1
	Khordha	1	Rasthali (AAB)	15	0125	R1
	Ranapur	1	Rasthali (AAB)	22	0125	R1
	Dhenkanal	1	Monthan (ABB)	17	01220, 0124	R1
	Kendrapara	1	Monthan (ABB)	9	0124	R1
	Jaipur	1	Monthan (ABB)	14	0124	R1
Maharashtra	Jalgaon	6	Grand Nain (AAA)	2–30	0125, 01213/16	R1, TR4
Sub-Tropical regions						
Gujarat	Surat	10	Grand Nain (AAA)	30–60	01220, 0120, 01213/16	R1, TR4, STR4
	Gandevi	1	Grand Nain (AAA)	5	0124, 0125	R1
	Bharuch	10	Grand Nain (AAA)	10–50	01213/16	TR4
	Vadodara	5	Grand Nain (AAA)	5–25	01220	R1
Uttar Pradesh	SantKabir Nagar	2	Monthan (ABB)	25 to 30	01220	R1
	Maharajganj	10	Grand Nain (AAA)	15 to 75	01216, 0125	TR4, R1
	Kushinagar	2	Grand Nain (AAA)	2	01220, 01213/16	TR4, R1
	Shravasti	6	Grand Nain (AAA)	2.5 to 60	01213/16	TR4
	Ayodhya	3	Grand Nain (AAA)	5 to 45	0125, 01220	TR4, R1
	Lakhimpur Kheri	10	Grand Nain (AAA)	7–90	01213/16	TR4
	Barabanki	2	Grand Nain (AAA)	1–20	01213/16	TR4
Madhya Pradesh	Burhanpur	6	Grand Nain (AAA)	10–50	0120, 01213/16	STR4, TR4
Bihar	Vaishali	8	Alpon (AAB)	1	0125	R1
		2	Chinia (Awak-ABB)	0.5–8	0124	R1
		1	Kothia (ABB)	0.5	01220	R1
		1	Malbhog (AAB)	1	0125	R1
		1	Muthia (ABB	0.5	01220	R1
		7	Grand Nain (AAA)	-	-	
	Khagaria		Chinia (ABB)	0.5	0124	R1
	Bhagalpur		Chinia (ABB)	0.1	0124	R1
			Robusta (AAA)	2	01213/16	TR4
	Katihar	7	Robusta (AAA)	2–27	01213/16	TR4
		9	Grand Nain (AAA)	6–65	01216	TR4
	Purnia	10	Chinia (ABB)	0.5	01213/16	TR4
			Robusta (AAA)	2–5	01213/16	TR4
			Grand Nain (AAA)	2 to 22.7	01213/16	TR4
West Bengal	Nadia	13	Grand Nain	0.5 to 50	0125, 01213/16	TR4, R1
		1	Kanthali (ABB)	40	0124, 01213/16	TR4, R1
		2	Mortaman (AAB)	3	0125	R1
		11	Singapuri (AAA)	6 to 50	01213/16	TR4
		1	Chinia (ABB)	10	0124	R1
		2	Cheni Champa (AAB)	3–33	01213/16	TR4
	Murshidabad	5	Singapuri (AAA)	12–26	01213/16	TR4
		1	Cheni Champa (AAB)	50	01213/16	TR4
	Hooghly	5	Kanthali (ABB)	0–20	0124	R1
	SJalpaiguri	1	Chinia (ABB)	1	0124	R1
Assam	Jorhat	6	Malbhog (AAB)	15–45	0125, 124, 0128, 0124/5, 01218, 01220, 01212, 0129	R1, STR4
		2	Monthan (ABB)	20–28	0125, 0124/5	R1
	Goalpara	4	Malbhog (AAB)	15–45	0125	R1
	Guwahati	1	Karpuravalli (ABB)	17	01211, 0124, 01220, 0125	STR4, R1
		2	Athiakol (BB)	0.1	0124/5, 0125	R1
Arunachal Pradesh	West Kameng	1	Karpuravalli (ABB)	0.5	0124	R1
		1	Athiakol (BB)	0.1	0124/5	R1
		3	Manohar (ABB)	0.1	0125, 0124	R1
Meghalaya	East Kashi Hills	2	Manohar (ABB)	2	0125, 0124	R1
Tripura	West Tripura	5	Sabri (Rasthali)	5–23	0125	R1
Nagaland	Dimapur	2	Manohar (ABB)	0.5	0125	R1

Pisang Awak (ABB)—Syn-Karpuravalli, Chinia, Manohar, Kanthali; Rasthali (Silk-AAB) Syn-Malbhog, Sabri, Amritapani, Nanjangud Rasabale, Mortaman; Cavendish (AAA)-Grand Nain, Robusta, Singapuri; Monthan (Cooking banana-ABB) Syn-Kothia, Muthia; Poovan-Mysore (AAB) Syn-Cheni champa, Alpon.

**Table 2 jof-10-00887-t002:** List of *Fusarium oxysporum* f. sp. *cubense* race-specific primers and *SIX* Primers used.

Primer Name	Primer Sequence (5′ to 3′)	Length (bp)	Annealing Conditions	Reference
*Foc*R1F *Foc*R1R	TACCTCCTTGGTCGACAGGT CAGACTTCCAACGTCTCGGT	320	62 °C	Thangavelu et al., 2022 [15]
*Foc*R4F *Foc*R4R	CGCACTCTTACGTTGAGGAT TCCACGCAACACTAGCTACT	400	66 °C	
*Foc*TR4F *Foc*TR4R	TGATTTGCCGTGGAATGACA TGGTCTTGACACGACCCA	250	65 °C	
*Foc*STR4F *Foc*STR4R	GCGCAAGTAGTCTTGCTTCC ATTAAGCGGTTGGCGTATTG	250	58 °C	
*SIX1a F* *SIX1a R*	GGCAAATCACTCGTCTGGGA CATAGCGGTAAAAGCCGCAC	573	60 °C	Raman et al., 2021 [20]
*SIX2a F* *SIX2a R*	TTTAGCACCGCGAGGAACTT AAACCAGCCACCATAACCGT	214	60 °C	
*SIX4a F* *SIX4a R*	GGCGTTGCGGGTTTTTAACT ATGCATTACGGAGTAGGCCC	717	60 °C	
*SIX6a F* *SIX6a R*	CTACGTCGACATCACTCCCA TACCATCATCTGCATCGCCA	412	59 °C	
*SIX7a F* *SIX7a R*	CCTCCTTTTCCATTTCGCCC CATTGGGCCTAAAGACGTCG	365	59 °C	
*SIX8a F* *SIX8a R*	CTTCCTCCTAGCCGTCTCTG TAAGCTCTTCACCTCACCCG	450	59 °C	
*SIX9a F* *SIX9a R*	ACATTCTGTCCGTCGATCGTT GCCACCTTCCATATCGCTGAA	275	59 °C	
*SIX13a F* *SIX13a R*	AACTCAAAGCCTCCTAGGCAC AATCCCTGGCGCTGACTTAG	332	60 °C	

**Table 3 jof-10-00887-t003:** Number of *Fusarium oxysporum* isolates collected from bananas in India.

State	Banana Cultivars							Number of Isolates
	Cavendish	Rasthali	Poovan	Monthan	Karpuravalli	NeyPoovan	Others	
Tamil Nadu	10	18	0	19	5	11	0	63
Pondicherry	0	0	0	0	2	0	0	2
Kerala	0	6	0	0	2	1	0	9
Karnataka	2	4	0	0	0	5	0	11
Andhra Pradesh	0	4	0	0	1	0	0	5
Odisha	0	3	0	3	0	0	0	6
Maharashtra	6	0	0	0	0	0	0	6
Gujarat	26	0	0	0	0	0	0	26
Uttar Pradesh	33	0	0	2	0	0	0	35
Madhya Pradesh	6	0	0	0	0	0	0	6
Bihar	38	1	8	2	5	0	0	54
West Bengal	29	2	2	0	8	0	0	41
Assam	-	10	0	2	1	-	2	15
Arunachal Pradesh	0	0	0	0	4	0	1	5
Meghalaya	0	0	0	0	2	0	0	2
Tripura	0	5	0	0	0	0	0	5
Nagaland	0	0	0	0	2	0	0	2
Total	150 (51.2)	53 (18.08)	10 (3.4)	28 (9.52)	32 (10.88)	17 (5.78)	3 (1.02)	293

**Table 4 jof-10-00887-t004:** The *Foc* VCGS/races affecting different banana varieties in tropical and subtropical regions of India.

States	Subgroup (Genome)	Varieties	Incidence	VCGs	Races
Tropical	Cavendish (AAA)	Grand Nain	2–95	0125, 0124, 01220	R1
	Silk (AAB)	Rasthali	0.4 to 45	0125, 0124/5, 0124, 0129, 01214, 01212, 01217	R1, STR4
	Cooking type (ABB)	Monthan	3–30	0125, 01220, 0128, 0124, 0124/5, 01212	R1
	Pisang Awak (ABB)	Karpuravalli	0.5 to 13	0124, 0125, 0128, 01220, 0124/5	R1
	NeyPoovan (AB)	NeyPoovan	1–30	0125, 0124/5, 0124, 01220, 01212, 01211	R1, STR4
Subtropical	Cavendish (AAA)	Grand Nain	0.5 to 95	01213/16, 01216, 0125, 01220, 0120	R1, STR4, TR4
	Mysuru (AAB)	Poovan	1–50	0125, 01213/16,	R1, TR4
	Pisang Awak (ABB)	Karpuravalli	0–20	0124, 01211, 01220, 0125, 01213/16	R1, STR4, TR4
	Cooking type (ABB)	Monthan	0.5 to 28	01220, 0125, 0124/5	R1
	Silk (AAB)	Rasthali	1–45	0125, 0124, 0128, 0124/5, 01218, 01220, 01212, 0129	R1, STR4 TR4
	Cavendish (AAA)	Robusta	6–50	01213/16	TR4
	Athiakol Type (BB)	Athiakol	0.1	0124/5, 0125	R1

**Table 5 jof-10-00887-t005:** *Fusarium oxysporum* f. sp. *cubense* isolates analysed for vegetative compatibility group (VCG) identity, pathogenicity to Cavendish cv. Grand Nain, and race.

S. No.	Isolate	Cultivar	Geographical Location	Race	VCG	Pathogenic	PCR Diagnosis for			
							TR4	STR4	R4	R1
1.	BR3	Monthan	Katihar, Bihar	1	01220	+	−	−	−	+
2.	BR5	Monthan	Katihar, Bihar	1	01220	+	−	−	−	+
3.	BR12	Grand Nain	Katihar, Bihar	4	01216	+	+	−	+	−
4.	BR13	Grand Nain	Katihar, Bihar	4	01213/16	+	+	−	+	−
5.	BR14	Grand Nain	Katihar, Bihar	4	01213/16	+	+	−	+	−
6.	BR19	Grand Nain	Katihar, Bihar	1	0124	+	−	−	−	+
7.	GU2	Grand Nain	Surat, Gujarat	1	01220	+	−	−	−	+
8.	GU3	Grand Nain	Surat, Gujarat	4	0120	+	−	+	+	−
9.	GU4	Grand Nain	Bharuch, Gujarat	4	01213/16	+	+	−	+	−
10.	GU5	Grand Nain	Bharuch, Gujarat	4	01213/16	+	+	−	+	−
11.	GU6	Grand Nain	Bharuch, Gujarat	4	01213/16	+	+	−	+	−
12.	GU14	Grand Nain	Vadodara, Gujarat	1	01220	+	−	−	−	+
13.	GU15	Grand Nain	Vadodara, Gujarat	1	01220	+	−	−	−	+
14.	GU21	Grand Nain	Surat, Gujarat	4	01213/16	+	+	−	+	−
15.	KA1	Grand Nain	Mandya, Karnataka	1	0124	+	−	−	−	+
16.	KA2	Monthan	Mandya, Karnataka	1	0124	+	−	−	−	+
17.	KA4	Nanjangudu Rasbale	Mysore, Karnataka	1	0125	+	−	−	−	+
18.	KA6	Grand Nain	Mysore, Karnataka	1	01220	+	−	−	−	+
19.	KA8	Grand Nain	Mysore, Karnataka	1	01220	+	−	−	−	+
20.	KE1	Rasthali	Thrissur, Kerala	1	0124	+	−	−	−	+
21.	KE2	Rasthali	Thrissur, Kerala	1	0125	+	−	−	−	+
22.	KE4	Big ebanga	Thrissur, Kerala	1	0124/5	+	−	−	−	+
23.	MP1	Grand Nain	Burhanpur, MP	1	0125	+	−	−	−	+
24.	MP2	Grand Nain	Burhanpur, MP	1	0125	+	−	−	−	+
25.	MP3	Grand Nain	Burhanpur, MP	1	01220	+	−	−	−	+
26.	MP4	Grand Nain	Burhanpur, MP	4	0120	+	−	+	+	−
27.	MA1	Grand Nain	Jalgaon, Maharashtra	1	0125	+	−	−	−	+
28.	TN1	Grand Nain	Theni, TN	1	0125	+	−	−	−	+
29.	TN2	Grand Nain	Theni, TN	1	0125	+	−	−	−	+
30.	TN3	Grand Nain	Theni, TN	1	0125	+	−	−	−	+
31.	UP8	Monthan	SantKabir Nagar, UP	1	01220	+	−	−	−	+
32.	UP11	Grand Nain	Maharajganj, UP	1	0125	+	−	−	−	+
33.	UP17	Grand Nain	Ayodhya, UP	4	01213/16	+	+	−	+	−
34.	UP20	Grand Nain	Maharajganj, UP	4	01213/16	+	+	−	+	−
35.	UP22	Grand Nain	Maharajganj, UP	4	01213/16	+	+	−	+	−
36.	UP40	Grand Nain	Shravasti, UP	4	01213/16	+	+	−	+	−
37.	UP41	Grand Nain	Barabanki, UP	4	01213/16	+	+	−	+	−
38.	WB2	Grand Nain	Nadia, WB	4	01213/16	+	+	−	+	−
39.	WB3	Grand Nain	Nadia, WB	1	0124	+	−	−	−	+
40.	WB7	Kanthali	Nadia, WB	4	01213/16	+	+	−	+	−
41.	WB8	Singapuri	Nadia, WB	4	01213/16	+	+	−	+	−
42.	WB9	Chinia	Nadia, WB	4	01213/16	+	+	−	+	−
43.	WB24	Singapuri	Murshidabad, WB	4	01213/16	+	+	−	+	−
44.	WB28	Kanthali	Nadia, WB	1	0125	+	−	−	−	+
45.	WB31	Kanthali	Hoogly, WB	4	01213/16	+	+	−	+	−
46.	WB32	Chinia	Hoogly, WB	4	01213/16	+	+	−	+	−

+ denotes present, − denotes absent.

**Table 6 jof-10-00887-t006:** Distribution of effector-based *SIX* genes in different strains of *Fusarium oxysporum* f. sp. *cubense* in India.

S. No.	Isolates	Cultivar	VCG	Races	*SIX1*	*SIX2*	*SIX4*	*SIX6*	*SIX7*	*SIX8*	*SIX9*	*SIX13*
1.	BR3	Monthan	01220	R1	+	−	+	+	−	−	+	+
2.	BR5	Monthan	01220	R1	+	−	+	+	−	−	+	+
3.	BR12	Grand Nain	01216	TR4	+	+	−	−	−	+	+	+
4.	BR13	Grand Nain	01213/16	TR4	+	+	−	−	−	+	+	+
5.	BR14	Grand Nain	01213/16	TR4	+	+	−	−	−	+	+	+
6.	BR19	Grand Nain	0124	R1	+	−	+	+	−	−	+	+
7.	GU2	Grand Nain	01220	R1	+	−	+	+	−	−	+	+
8.	GU3	Grand Nain	0120	STR4	+	+	−	−	+	+	+	+
9.	GU4	Grand Nain	01213/16	TR4	+	+	−	−	−	+	+	+
10.	GU5	Grand Nain	01213/16	TR4	+	+	−	−	−	+	+	+
11.	GU6	Grand Nain	01213/16	TR4	+	+	−	−	−	+	+	+
12.	GU14	Grand Nain	01220	R1	+	−	+	+	−	−	+	+
13.	GU15	Grand Nain	01220	R1	+	−	+	+	−	−	+	+
14.	GU21	Grand Nain	01213/16	TR4	+	+	−	−	−	+	+	+
15.	KA1	Grand Nain	0124	R1	+	−	+	+	−	−	+	+
16.	KA2	Monthan	0124	R1	+	−	+	+	−	−	+	+
17.	KA4	Nanjangudu Rasbale	0125	R1	+	−	+	+	−	−	+	+
18.	KA6	Grand Nain	01220	R1	+	−	+	+	−	−	+	+
19.	KA8	Grand Nain	01220	R1	+	−	+	+	−	−	+	+
20.	KE1	Rasthali	0124	R1	+	−	+	+	−	−	+	+
21.	KE2	Rasthali	0125	R1	+	−	+	+	−	−	+	+
22.	KE4	Big ebanga	0124/5	R1	+	−	+	+	−	−	+	+
23.	MP1	Grand Nain	0125	R1	+	−	+	+	−	−	+	+
24.	MP2	Grand Nain	0125	R1	+	−	+	+	−	−	+	+
25.	MP3	Grand Nain	01220	R1	+	−	+	+	−	−	+	+
26.	MP4	Grand Nain	0120	STR4	+	+	−	−	+	+	+	+
27.	MA1	Grand Nain	0125	R1	+	−	+	+	−	−	+	+
28.	TN1	Grand Nain	0125	R1	+	−	+	+	−	−	+	+
29.	TN2	Grand Nain	0125	R1	+	−	+	+	−	−	+	+
30.	TN3	Grand Nain	0125	R1	+	−	+	+	−	−	+	+
31.	UP8	Monthan	01220	R1	+	−	+	+	−	−	+	+
32.	UP11	Grand Nain	0125	R1	+	−	+	+	−	−	+	+
33.	UP17	Grand Nain	01213/16	TR4	+	+	−	−	−	+	+	+
34.	UP20	Grand Nain	01213/16	TR4	+	+	−	−	−	+	+	+
35.	UP22	Grand Nain	01213/16	TR4	+	+	−	−	−	+	+	+
36.	UP40	Grand Nain	01213/16	TR4	+	+	−	−	−	+	+	+
37.	UP41	Grand Nain	01213/16	TR4	+	+	−	−	−	+	+	+
38.	WB2	Grand Nain	01213/16	TR4	+	+	−	−	−	+	+	+
39.	WB3	Grand Nain	0124	R1	+	−	+	+	−	−	+	+
40.	WB7	Kanthali	01213/16	TR4	+	+	−	−	−	+	+	+
41.	WB8	Singapuri	01213/16	TR4	+	+	−	−	−	+	+	+
42.	WB9	Chinia	01213/16	TR4	+	+	−	−	−	+	+	+
43.	WB24	Singapuri	01213/16	TR4	+	+	−	−	−	+	+	+
44.	WB28	Kanthali	0125	R1	+	−	+	+	−	−	+	+
45.	WB31	Kanthali	01213/16	TR4	+	+	−	−	−	+	+	+
46.	WB32	Chinia	01213/16	TR4	+	+	−	−	−	+	+	+

+ denotes present, − denotes absent.

## Data Availability

The TEF 1 α sequences of *Foc* isolates used for genome BLAST and phylogenetic analysis have been deposited in the NCBI data base. All the data generated or analysed during this study are included in this published article.

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
