# Peer review of "Geographical Distribution, Host Range and Genetic Diversity of Fusarium oxysporum f. sp. cubense Causing Fusarium Wilt of Banana in India"

_jof, 2024, doi:10.3390/jof10120887_

Round 1

Reviewer 1 Report

This study conducted field surveys in major banana growing regions of India to gain insight in the extend of spread of the disease and to know the distribution of different Foc races and VCGs of Fusarium wilt of banana in India. It contains important information to be published, but there are several points to be improved before publication.

1. Title. The names of the genus and the species should be should be written in italics. Please check the full text.

2. Abstract. The abbreviation should be adjusted the order of occurrence. For example, Line 30 &33. Secreted in Xylem(SIX).

3.Introduction. Line 55-74. Please add the related references.

4. What does CLA refer to ?

5. It is far from enough to identify Fusarium only with TEF- 1α gene. It is recommended to add genes or please explain why TEF- 1α gene was chosen.

6. Figure 4. The outgroup species should be two or two more samples. I suggest that the tree is rooted with two samples of Fusarium equiseti or two species, so that the support rate is higher.

7. Discussion. Paragraph 1&2. Please simplify, or merge into the introduction section.

8. References. Line 564, 577, 593, 625, 633, 639, 644. Please check the format.

Author Response

Response to the reviewers comments

Reviewer 1

  1. Comment: The names of the genus and the species should be should be written in italics. Please check the full text.

Response: Agreed. The Genus and species were changed to italics in the whole text

  1. Comment: Abstract. The abbreviation should be adjusted the order of occurrence. For example, Line 30 &33. Secreted in Xylem (SIX).

Response:  Agreed and carried out

  1. Comment Introduction. Line 55-74. Please add the related references.

Response:  For this, source of information is mentioned in the text

  1. Comment What does CLA refer to ?

Response:  Carnation leaf agar (CLA) and the same is mentioned in the MS

  1. Comment It is far from enough to identify Fusariumonly with TEF- 1α gene. It is recommended to add genes or please explain why TEF- 1α gene was chosen.

Response:  The translation elongation factor 1-alpha (TEF1-α) gene is important in the phylogenetic analysis of Foc because it is a highly conserved protein that is a key part of the protein translation machinery. This is being used universally for a) Species identification in Fusarium b) Sequence polymorphism among related species c) Reduces incorrect phylogenetic inferences and d) The TEF nucleotide sequence is a powerful marker that shows associations among fungi at all levels. 

In addition to TEF1-α gene sequences, In this present study, we have carried out other important methods of Identifications such as VCG analyses, Foc race specific markers, Blastn analyses and the morphological identification 

  1. Comment Figure 4. The outgroup species should be two or two more samples. I suggest that the tree is rooted with two samples of Fusarium equisetior two species, so that the support rate is higher.

Response:  Agreed. Included two isolates of F. equiseti as an out group for the phylogenetic analyses

  1. Comment Discussion. Paragraph 1&2. Please simplify, or merge into the introduction section.

Response:  Agreed. Simplified the para 1 and 2 in the discussion

  1. Comment References. Line 564, 577, 593, 625, 633, 639, 644. Please check the format.

Response:  Agreed. Checked and corrected the mistakes

Reviewer 2 Report

The manuscript entitled “Geographical distribution, host range and genetic diversity of Fusarium oxysporum f. sp. cubense causing Fusarium wilt of banana in India” is a very interesting research on the study of Fusarium genera in India; some points are recommended to the authors:

1.       In the Materials and Methods section, does the figure 297 correspond to the number of plantations visited, as indicated in Table 1? The numbers do not match. Additionally, revise the phrasing of the sentence: "A roving survey was conducted on 297 farms in 53 districts and 17 banana-growing states of India were conducted from 2019 to 2023 to assess the incidence of banana Fusarium wilt and to collect Foc samples for the isolation and characterization of the pathogen (Table 1)" A map reflecting this information should be created or included in the map of Figure 3.

2.       In the Results section, section 3.1 does not reflect the origin of the samples and isolates from the total of 297 farms across the 17 states mentioned (this information is also missing in Figure 3, which only indicates 15 states). It is also unclear why 293 isolates were obtained from 732 samples.

3.       Section 3.2 presents very general data; the authors should provide more details in this section. Figure 2 is repetitive of the text. It is recommended to enhance the figure by including more information that reflects the variation of VCGs in different banana cultivars. In general, Figure 3 does not adequately represent the study, it is recommended to improve it.

4.       In the Discussion section, do the authors provide an explanation for how the geographic distribution of Foc in India was determined?

Detail comments:

1.       In the Authors and Institutions section, adjust the email addresses to correspond to the respective institutions. Institutions 2 and 3 are missing their country designation.

2.       In the Abstract, add the full meaning of the abbreviation IDM.

3.       In the Introduction, all internet links are non-functional. It is recommended to follow the journal's guidelines for citing websites or use the most appropriate type of reference. The same recommendation applies to "Personal observation" and "unpublished data."

4.       In section 2.4, define the term "3-mo-old potted TC" and explain how the five plants were inoculated with each isolate (line 163). In section 2.1, interviews with farmers are mentioned, but only in very general terms. The results are addressed in the discussion. It is recommended to reconsider the presentation of this information.

5.       In section 2.6, define the term "3-mo-old". In section 2.7.1, indicate the modifications made to the DNA extraction method. In section 2.7.2, correct the "-1" units by formatting them as subscripts, and use the proper degree symbol for Celsius (also applies to section 2.7.3.).

6.       Consider merging Tables 3 and 4 and include the corresponding references for the primers used. In section 2.7.3, clarify how the 46 isolates evaluated were selected. In section 2.7.4, specify which isolates were analyzed.

7.       Review the manuscript for spacing errors. For example: "isolatedfrom," "tissueswere," "applying30," "5and," "was96.8," "India.AFusarium," "levelincidence," "30].In.", etc. Add spaces where needed.

8.       Place tables and figures as close as possible to the text where they are cited.

9.       Supplementary tables are not included and used in the manuscript.

Author Response

Response to the reviewers comments

Reviewer -2

The manuscript entitled “Geographical distribution, host range and genetic diversity of Fusarium oxysporum f. sp. cubense causing Fusarium wilt of banana in India” is very interesting research on the study of Fusarium genera in India; some points are recommended to the authors:

  1. CommentIn the Materials and Methods section, does the figure 297 correspond to the number of plantations visited, as indicated in Table 1? The numbers do not match. Additionally, revise the phrasing of the sentence: "A roving survey was conducted on 297 farms in 53 districts and 17 banana-growing states of India were conducted from 2019 to 2023 to assess the incidence of banana Fusarium wilt and to collect Focsamples for the isolation and characterization of the pathogen (Table 1)" A map reflecting this information should be created or included in the map of Figure 3.

Response : Thank you for pinpointing the mistakes. Corrected the same as 286. Phrase is revised. As suggested, the information of number of farm visited is included in Fig 3

  1. CommentIn the Results section, section 3.1 does not reflect the origin of the samples and isolates from the total of 297 farms across the 17 states mentioned (this information is also missing in Figure 3, which only indicates 15 states). It is also unclear why 293 isolates were obtained from 732 samples.

Response Agreed and corrected in Fig 3 (16 States and I union territory). We collected a minimum of two to three plant samples (732 samples) from Foc affected plant of each Foc affected farm. After isolation of Foc and carrying out the pathogenicity study, we have taken a representative sample of 293 representing all the farm affected with Foc and carried out VCG and molecular methods for characterization and identification of Foc races.

  1. Comment:   Section 3.2 presents very general data; the authors should provide more details in this section. Figure 2 is repetitive of the text. It is recommended to enhance the figure by including more information that reflects the variation of VCGs in different banana cultivars. In general, Figure 3 does not adequately represent the study, it is recommended to improve it.

Response: Agreed. As suggested, the details are given for the pathogenicity study and in addition more details are given in Fig 2 and 3

  1. CommentIn the Discussion section, do the authors provide an explanation for how the geographic distribution of Foc in India was determined?

Response : This has already been provided in the discussion part line number 418 to 427

Detailed comments:

  1. Comment:In the Authors and Institutions section, adjust the email addresses to correspond to the respective institutions. Institutions 2 and 3 are missing their country designation.

Response: Mistakes are rectified

  1. Comment:In the Abstract, add the full meaning of the abbreviation IDM.

Response: Yes added

  1. Comment:In the Introduction, all internet links are non-functional. It is recommended to follow the journal's guidelines for citing websites or use the most appropriate type of reference. The same recommendation applies to "Personal observation" and "unpublished data."

Response: The internet links are made in to functional. Yes, for all the references the journal guidelines are followed.  In the case of comments on "Personal observation" and "unpublished data I don’t know how to give citation.

  1. Comment:     In section 2.4, define the term "3-mo-old potted TC" and explain how the five plants were inoculated with each isolate (line 163). In section 2.1, interviews with farmers are mentioned, but only in very general terms. The results are addressed in the discussion. It is recommended to reconsider the presentation of this information.

Response: It is the 3-month-old potted plants and the same is mentioned in the text. Other details are given elaborately in materials and methods. The information related to pathogenicity is given in results section. 

5.. Comment: In section 2.6, define the term "3-mo-old". In section 2.7.1, indicate the modifications made to the DNA extraction method. In section 2.7.2, correct the "-1" units by formatting them as subscripts, and use the proper degree symbol for Celsius (also applies to section 2.7.3.).

Response: I section 2.6 it is the 3-month-old plants. In 2.7.1. the modifications made were i) instead of two steps of addition of phenol followed by chloroform isoamyl alcohol we have added all together in a single step. ii) For precipitation of DNA, added both isopropanol and 3M sodium acetate instead of isopropanol alone.

  1. Comment: Consider merging Tables 3 and 4 and include the corresponding references for the primers used. In section 2.7.3, clarify how the 46 isolates evaluated were selected. In section 2.7.4, specify which isolates were analyzed.

Response: As suggested table 3 and 4 are merged. For the selection of 46 isolates the clarification is given in the text line number 238 to 242. The details of isolates analyzed are given in table 6 (the 46 isolates used for both TEF1 and SIX gene analyses are the same). 

  1. Comment: Review the manuscript for spacing errors. For example: "isolatedfrom," "tissueswere," "applying30," "5and," "was96.8," "India.AFusarium," "levelincidence," "30].In.", etc. Add spaces where needed.

Response: All typos’ errors are corrected

  1. Comment: Place tables and figures as close as possible to the text where they are cited.

Response: Agreed. Carried out as suggested

  1. Comment:Supplementary tables are not included and used in the manuscript.

Response:  Supplementary figures are included in the main MS

Reviewer 3 Report

The study presented a detailed context of the Fusarium wilt epidemic progress in the banana cropping system in India 

please double check for typos along the manuscript 

what was the main outcome of the crossing infections in the plant material testing and how this analysis supports the identification of R1 infecting cavendish type plants

Please provide data on the pathogenicity scale implemented

Typos in section 2.3

 Provide more detailed explanation on the analysis of SIX genes in Ln 513-519, what SIX1 gene you are mention a, b or c type from TR4? 

Author Response

Response to the reviewers comments

Reviewer 3

The study presented a detailed context of the Fusarium wilt epidemic progress in the banana cropping system in India 

1 Comment: please double check for typos along the manuscript 

Response: Yes, checked and corrected

  1. Comment: what was the main outcome of the crossing infections in the plant material testing and how this analysis supports the identification of R1 infecting cavendish type plants

Response: The cross-infection studies carried out in cv. Grand Nain (Cavendish AAA) using the dominant VCGs identified in India Viz. 0124, 0125, 0124/5, 01220 of Foc R1 were able to infect the Grand Niane and cause 3-5 internal wilt score under the glass house condition.

  1. Comment: Please provide data on the pathogenicity scale implemented

Response: In materials and methods reference is already given.

Detail comments

  1. Comment: Typos in section 2.3

 Response: Yes, carried out

  1. Comment: Provide more detailed explanation on the analysis of SIX genes in Ln 513-519, what SIX1 gene you are mentioning a, b or c type from TR4? 

Response: The detailed explanation about the SIX genes analyses is given in section 3.5.3 and table 6. We have designed primer for the for the SIX 1 gene and not for the sequence variation of SIX 1 gene. Hence the results provided are for SIX gene only.